# Leveraging Adversarial Examples to Obtain Robust Second-Order Representations

## Abstract

Deep neural networks represent data as projections on trained weights in a high dimensional manifold. This is a first-order based absolute representation that is widely used due to its interpretable nature and simple mathematical functionality. However, in the application of visual recognition, first-order representations trained on pristine images have shown a vulnerability to distortions. Visual distortions including imaging acquisition errors and challenging environmental conditions like blur, exposure, snow and frost cause incorrect classification in first-order neural nets. To eliminate vulnerabilities under such distortions, we propose representing data points by their relative positioning in a high dimensional manifold instead of their absolute positions. Such a positioning scheme is based on a data point's second-order property. We obtain a data point's second-order representation by creating adversarial examples to all possible decision boundaries and tracking the movement of corresponding boundaries. We compare our representation against first-order methods and show that there is an increase of more than $14\%$ under severe distortions for ResNet-18. We test the generalizability of the proposed representation on larger networks and on 19 complex and real-world distortions from CIFAR-10-C. Furthermore, we show how our proposed representation can be used as a *plug-in approach* on top of any network. We also provide methodologies to scale our proposed representation to larger datasets.

## 1 Introduction

In recent years, artificial intelligence systems achieved state-of-the-art performances in image classification tasks (Russakovsky et al., 2015)(Krizhevsky et al., 2012)(He et al., 2016). Specifically, classification algorithms surpassed top-5 human error rate of $5.1\%$ on ImageNet (Russakovsky et al., 2015). Even though these advancements are promising, images in these datasets do not cover diverse real-world scenarios. For instance, ImageNet consists of photographs parsed from Flickr, a popular image hosting service. The images on Flickr are generally high quality since users tend not to share distorted photographs. Distortions may include perceptually unpleasant camera related issues like blur, motion blur, overexposure, underexposure, and noise. Moreover, environmental conditions such as rain, snow, and frost can affect the field of view. These non-ideal conditions impact the performance of artificial intelligence (AI) algorithms (Dodge & Karam, 2017)(Hendrycks & Dietterich, 2019).

These AI algorithms are primarily driven by deep neural nets that learn non-linear transformations to obtain discriminate representation spaces. Deep neural networks are trained to transform a data point into a representation space where linear classifiers can discriminate between classes (Goodfellow et al., 2016)(Krizhevsky et al., 2012)(He et al., 2016). All the hidden layers are supervised to maximize the interclass distance while minimizing the intraclass distance to obtain linearly separable representations in the last layer. We formulate the inherent mechanisms behind the classification process as follows: Let $f$ be an $L$ layered neural network trained to distinguish between $N$ classes. If $x$ is any input to the network, the output for a classification application is given by $f(x) = y$ where $y$ is a $(N \times 1)$ vector. The class of $x$ is the index of the maxima of $y$. Consider only the final fully connected layer $f_L$ parameterized by weights ($W_L$) and bias ($b_L$). We obtain $y$ as,

$$y = W_L^T f_{L-1}(x) + b_L,$$
$$\forall y \in \Re^{N \times 1}, W_L \in \Re^{d_{L-1} \times N}, f_{L-1}(x) \in \Re^{d_{L-1} \times 1}, b_L \in \Re^{N \times 1}, \tag{1}$$

where $f_{L-1}$ is the flattened output of the network just before the final fully connected layer. All the data points that span $f_{L-1}$ representation space should be linearly separable for a well trained $f_L$. Note that Eq. 1 is a filtering operation between the weight vectors and the representation. Hence, the final fully connected layer in a network can be considered as a linear filter set with $N$ filters onto which the representations ($f_{L-1}$) are linearly projected (Wang et al., 2019). Projection refers to a dot product between the data point and the filters. The filter $W_L^i, \forall i \in [1, N]$ that has the largest projection or is maximally correlated with the data point represents the corresponding class.

In the above setting, the original data point $x$ is represented by its projection intensity $f_{L-1}(x)$. This representation can be analyzed as a first-order point process (Dorai-Raj et al., 2001). As an example, consider cities in the USA. All cities can be located on a map using their latitudes and longitudes as shown in Fig.1a. This is an instance of directly using the intensity - in this case the latitude and longitude - to represent a data point - in this case a city. The first-order representation is widely used for its intuitive nature, interpretability and ease of mathematical operation.

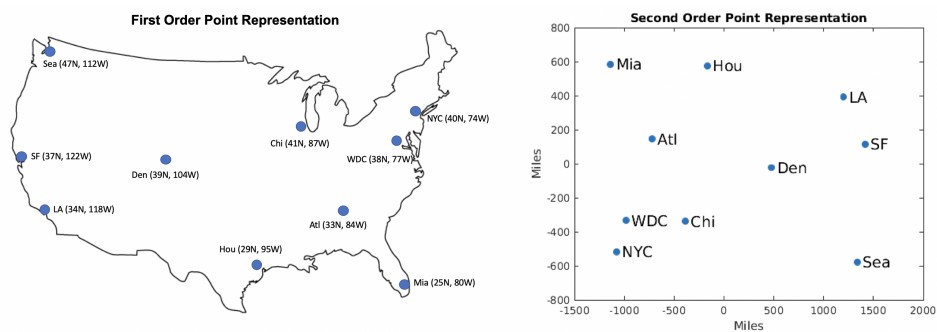

Figure 1: (a) Cities located using first order point representation - Latitudes and Longitudes. (b) Cities located based on second order point representation - Pairwise Distance.

However, under distortions and perturbations, this representation is not stable (Azulay & Weiss, 2018)(Goodfellow et al., 2014). The authors in Goodfellow et al. (2014) show the instability of such systems with adversarial images. In Azulay & Weiss (2018), the authors demonstrate that when images are translated by a few pixels, the network output and the hidden layer representations change drastically. We show that this change holds for distortions as well in Fig.2. Consider an image taken from MNIST (LeCun et al., 1998) dataset. The image is subjected to five levels of progressive blurring all of which are visualized in Fig.2a. Fig.2b shows the t-SNE (Maaten & Hinton, 2008) feature visualization of first order point representations of corresponding $f_{L-1}(x_{BlurLvl})$ for 10000 images in MNIST test set. The individual shapes and absolute locations of each cluster get progressively distorted with increasing levels of blur. Hence, it is natural that a network trained on original data performs poorly on distorted data.

An alternative to representing a point by its intensity is by representing its influence on every other point in the subspace (Dorai-Raj et al., 2001). This is the second-order property of data points. Consider the same example as before from Fig.1a. However, instead of latitudes and longitudes, the pairwise distance between all cities is provided. A Multi-Dimensional Scaling (MDS) algorithm is used to obtain the relative positioning of the cities with respect to each other as shown in Fig.1b. MDS algorithms (Mead, 1992) are a class of algorithms that use second-order point representations to translate pairwise distances of $N$ objects into an abstract cartesian space.

In Sections 2 and 3 , we provide an intuitive methodology to extract the second-order representation of any data point. In Section 4, we categorize the existing literature for classification under distortion and organize them into frameworks based on data dependency. We describe the experimental setup and discuss the results in Section 5 and conclude our work in Section 6.

## 2 SECOND ORDER POINT REPRESENTATIONS

Second-order point representations have been utilized in popular ML algorithms like k-Nearest Neighbor (kNN) classification. kNN classifies a data point based on the classes of k of its nearest neighbors, nearest being determined by a predefined distance metric. However, these algorithms are computationally expensive. Similarly, classical MDS algorithms (Hout et al., 2013)(Mead, 1992) on

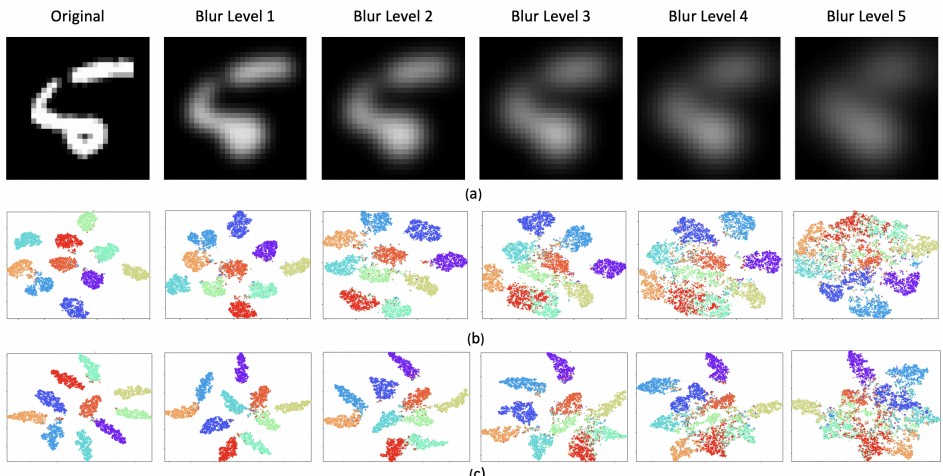

Figure 2: (a) Progressive blurring of a sample image from MNIST dataset (b) t-SNE maps of corresponding blurry MNIST testset for first-order representation (c) t-SNE maps of corresponding blurry MNIST testset for proposed second-order representation.

$M$ datapoints, require formation of a pairwise distance matrix of size $M \times M$, which is not feasible for large-scale datasets.

In this paper, we propose a proxy method for representing a data point using its second-order property. Instead of representing a data point as pairwise distance against other points, we represent it as a cumulative distance to all decision boundaries. We use adversarial image generation along with gradients of the final fully connected layer to obtain this cumulative distance. Consider Fig. 3a where the data point is projected onto a filter from the final fully connected layer, $W_L^i, \forall i \in [1, N]$. This provides a projection $P_i, \forall i \in [1, N]$. During backpropagation, if $i$ is the given class for data point $x$, the filter moves in a direction that maximizes $x$'s new projection $P_i'$. The movement of the filter is controlled by gradients. Hence, gradients provide the negative direction of a data point to a filter $W_L^i$. Our goal is to not just get the direction to the maximum projection class but to all the classes. Hence, by backpropagating over all $i \in [1, N]$ for a datapoint, we obtain the direction of that data point to every filter $W_L^i$. However, one iteration of backpropagation is not sufficient to obtain the distance between a datapoint to the decision boundary of a filter $W_L^i$. Hence, we generate a targeted adversarial image with class $i, \forall i \in [1, N]$ for the datapoint $x$ and absolute sum the gradients over the required iterations. This cumulative gradient provides the distance from a datapoint to a decision boundary. Creating adversarial images for every class $i \in [1, N]$ provides multidirectional features. This concept is shown in Fig 3b. We create $N$ adversarial images for every datapoint and track their individual movement until they hit a decision boundary.

These features form a proxy second-order representation for a datapoint. The benefits of this representation are illustrated in the t-SNE maps for the MNIST testset in Fig. 2c. The relative shapes of the class clusters are preserved for the proposed representation in Fig. 2c compared to first-order representations in Fig. 2b. Further stability results are discussed in Appendix A.1

## 3 MULTI-DIRECTIONAL FEATURE GENERATION

Continuing the notations established in Secs. 1 and 2, a $L$ layered network $f$ is trained to distinguish between $N$ classes using original distortion free images. For every image $x$ in the distortion free dataset, targeted Fast Gradient Sign Method (Goodfellow et al., 2014) (FGSM) is applied to create $N$ adversarial images to obtain $x$'s distance to decision boundaries. For a target $i \in [1, N]$, an adversarial noise $\epsilon \text{sign}(\nabla_x J(W, x, i))$ is added to the image $x$. $J(W, x, i)$ refers to the cost function used to train the network with parameters $W$. $\nabla_x$ is the gradient of the cost function w.r.t. the input $x$. $\epsilon$ of 0.1 is used in this work. Adversarial noise is added to the input over $k$ iterations until the classification changes to $i$, i.e $f(x_{k-1} + \epsilon \text{sign}(\nabla_x J(W, x_k, i))) = i$. The absolute value of the gradient of the cost function with respect to filter $W_L^i$ is summed up over $k$ iterations, i.e $r_i = \sum_{j=0}^{k-1} \text{abs}(\nabla_{W_L^i} J(W, x_j, i))$ where $r_i$ is the feature for $i^{\text{th}}$ class. $i$ is then iterated over all N

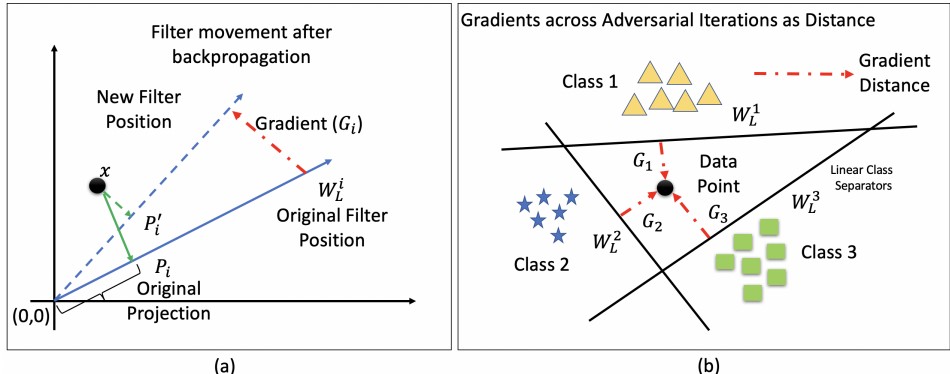

Figure 3: (a) Motivation for using gradients as directional information. (b) Motivation for constructing multi-target adversarial images.

classes to obtain multi-directional features of $x$ to all decision boundaries. All corresponding $r_i$ are concatenated to obtain the final feature $r_x = [r_1 r_2 \dots r_N]$. Hence the final multi-directional feature is given by,

$$r_x = \left[ \sum_{j=0}^{k-1} \text{abs}(\nabla_{W_L^1} J(W, x_j, 1)) \sum_{j=0}^{k-1} \text{abs}(\nabla_{W_L^2} J(W, x_j, 2)) \cdots \sum_{j=0}^{k-1} \text{abs}(\nabla_{W_L^N} J(W, x_j, N)) \right] \tag{2}$$

$r_x$ is the proposed proxy second-order representation of $x$. Note that if the dimensionality of the $(L-1)^{\text{th}}$ layer from Eq.1 is $d_{L-1} \times 1$, then the dimensionality of every subfeature $r_i$ is also $d_{L-1} \times 1$. The concatenated final feature has a dimensionality $(N * d_{L-1}) \times 1$. For reference, if the network $f_L$ is VGG-16 trained on CIFAR-10, $d_{L-1}$ is $512 \times 1$ and $r_x$ is $5120 \times 1$. This representation can get prohibitive with increased number of classes. Simple methods to offset this increase and speed up $r_x$ generation are discussed in Sec. 5.3.

The multi-directional features $r_x$ are as analogous to $f_{l-1}(x)$ as pairwise distances are to longitudes and latitudes in Fig. 1. Hence, $r_x$ is used in the same way as $f_{l-1}(x)$. However because of change in values and dimensionality, the same final fully connected layer $f_L$ cannot be used to discriminate $r_x$. Instead we train a new classifier $g_L$ on top of $r_x$. $g_L$ is a new network whose parameter dimensions depend on inputs. For instance, in Sec. 5.1, $g_L$ is a 3 layer fully connected network separated by ReLU non-linearity for ResNet-18 whose $r_x$ dimensions are $640 \times 1$. For ResNet-101 in Sec. 5.2, whose $r_x$ dimensions are $2560 \times 1$, $g_L$ is a 5 layered fully connected network. On a NVIDIA 1080TI GPU, $g_L$ for ResNet-101 trains within 320 secs.

The second-order representation of any data point is obtained from Eq.2. Note that the generation of gradients is a deterministic procedure and does not change over repeated experiments for the same network. However, the generation procedure is tied to a specific network and hence the representation of the same $x$ differs for different networks. In Section 5, we implement 5 networks and show that the proposed second order representation is robust compared to its first order counterpart in all of them. Moreover, the cost function $J(W, x, i)$ used to obtain gradients, impacts the value of said gradients and creates differences in $r_x$. In our experiments, we use MSE loss function to define $J$. Results comparing multiple loss functions on distorted CIFAR dataset is shown in Appendix A.2.

## 4   EXISTING FRAMEWORK AND RELATED WORKS

All the existing literature for alleviating the effects of distortions for classification use first-order representations to describe a data point. Moreover, a majority of the algorithms use some form of distorted data during training. The authors in Vasiljevic et al. (2016) show that finetuning VGG-16 using blurry training images increases the performance of classification under blurry conditions. However, performance between different types of blurs is not generalized and knowledge of type of blur is required for good performance. The authors in Zhou et al. (2017) propose finetuning and retraining early feature extraction layers of the network to increase classification accuracy. Temel et al.

(2017) propose utilizing distorted virtual images to boost performance accuracy. All these works require knowledge of distortion and large amounts of distorted data during training that makes them impractical in real life scenarios like in autonomous driving. We categorize methods that require distorted data while training into a Full-Reference (FR) framework. Retraining, finetuning and data augmentation techniques that utilize distorted data during training fall under the FR framework.

In contrast, any framework that does not require distorted images during training will be termed as No-Reference (NR) framework. The authors in Hendrycks & Dietterich (2019) show adversarial defense schemes can increase distortion robustness. Most of the existing NR approaches generally try to denoise (Buades et al., 2005) the distorted test image before feeding into a network. These include Hossain et al. (2018) where the authors use a DCT module to remove high frequency components from every image. Generally, these denoising algorithms are both computationally expensive and time consuming (Vasiljevic et al., 2016). Moreover, denoising algorithms require parameter tuning that varies by distortion types, levels and image resolutions. The proposed second order representation can be applied on top of any network, trained in either FR or NR fashion. Hence, by definition our approach follows a NR framework. Even when $f$ is trained in a FR framework, we obtain $r_x$ from undistorted $x$ and train $g_L$ in a NR fashion.

Additionally, we propose a new framework between the FR and NR frameworks called Reduced-Reference (RR). RR framework require only a few statistics of the distorted images for training $g_L$. RR framework differs from domain adaptation since it does not need test domain images during training but only their statistics. Such a framework is more appropriate in practical applications where it is impractical to anticipate, obtain or record all possible distortions. While the results of RR approaches may not exceed FR techniques, they come at a much cheaper data price. In our case, we convert our NR framework to RR by normalizing $r_x$ with mean and standard deviation statistics of distorted data. The terminologies of FR, NR, and RR are borrowed from the Image Quality Assessment community (Prabhushankar et al., 2017).

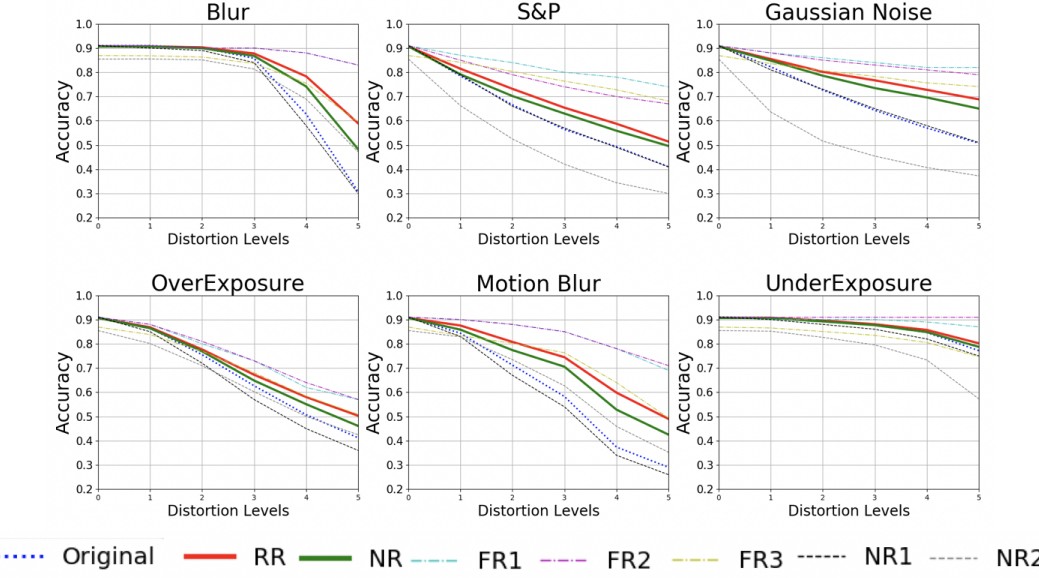

Figure 4: Results on CIFAR-10. Original is the first-order ResNet-18. NR is the proposed second-order representation and RR is a statistics enhanced proposed second-order representation.

## 5 RESULTS

The performance of second-order over first-order representations is validated in this section.

### 5.1 MULTI-DIRECTIONAL FEATURES VS FIRST-ORDER FEATURES

**Comparison against first order representation:** CIFAR-10 (Krizhevsky & Hinton, 2009) dataset is used to evaluate the effectiveness of the proposed second-order representation space against first-order methods. We consider six distortions - gaussian blur, salt and pepper, gaussian noise, overexposure,

motion blur, and underexposure. Five progressively worsening levels of all six distortions are created on CIFAR-10 dataset similar to the procedure followed by Temel et al. (2018). ResNet-18 is chosen as the original model $f$, trained on undistorted data. The accuracy on progressively worsening distortions for $f$ are visualized in Fig. 4 in blue where the y-axis shows the drop in performance and x-axis shows the corresponding distortion level. Level 0 indicates that there is no distortion. A significant performance drop is noticed across distortions for $f$. In contrast, when the original model's first-order representation is converted to proposed second-order, the results increase by an average of 5.21% across levels and distortions. This is shown in the green NR curve. The difference is more pronounced under level 5 severe distortions where the average accuracy across distortions increases by 10.04%. The red curve is the proposed RR second-order representation. The proposed RR outperforms the original by an average 7.72% across all distortions and levels. When there are no distortions, the level 0 results in the plots suggest that the proposed second-order representation performs similar to the first-order original representation.

**Comparison against NR models:** We compare the proposed second-order representation against 2 NR methods. NR1 technique denoises distorted test data by Non Local Means (NLM) before feeding into $f$. Explicit denoising by NLM (Buades et al., 2005) was analyzed by Hendrycks & Dietterich (2019). Because of the small resolution of CIFAR-10 images, NLM significantly blurs the distorted images, thereby adding additional artifacts into the already distorted images. Hence it performs worse than the original model under all distortions. Similar results were obtained using Hossain et al. (2018) where a drop of 8% was observed even on pristine images. NR2 is an adversarially robust classifier trained using undistorted images and untargeted FGSM images. Except in blur distortion, NR2 under performs the proposed NR in all other distortions performing even worse than the original model in Gaussian noise and underexposure.

**Comparison against FR models:** The original model's second-order performance gains are compared against 3 FR models - FR1 (Zhou et al., 2017), FR2 (Vasiljevic et al., 2016), and FR3, a data augmentation technique. The performance of the proposed RR and NR methods slot near the middle of the original and FR1 and FR2 models. The performance gain of proposed RR vs the original is 7.72% while the gain of FR1 over proposed RR is 7.70%. Note that both FR1 and FR2 require separate models for every distortion and distortion level i.e there are 31 trained models for each of FR1 and FR2 to produce the plots in Fig. 4. This shows that the proposed second-order representation can be a useful tool in practical applications to enhance the original model's performance where distorted images are not easily obtainable during training. FR3 is trained on distortion free images augmented with 1000 images each from all 6 distortions. It can be seen that the proposed RR representation performs similarly to FR3 in all but SP noise. Moreover, for pristine data at level 0, FR3 performs worse than the second-order representations.

**Comparison against other second order methods :** We compare our proxy second-order representation against true second-order representations. All existing networks utilize first-order representations to make decisions. To simulate a second-order scenario, We construct k-Nearest Neighbor (kNN) classifier on activations from $f_{L-1}(x)$. We do so for four $k$ values - 5, 25, 50, 500. Our proxy second-order NR outperforms all kNN classifiers by 6.37% averaged over all 6 distortions among all 5 levels on ResNet-18. This is because instead of calculating pairwise distance between simultaneously moving distorted data, we calculate distance between fixed (decision boundary) and one moving distorted data. Such a proxy method provides more robustness than its true second-order counterpart.

## 5.2 MULTI-DIRECTIONAL FEATURE ROBUSTNESS

**Robustness on multiple architectures:** The significance of the second-order representation derives from its broad utility and applicability as a "plug-in" on any gradient-based neural network. In Section 5.1, the proposed representation performs similar to FR3. But once FR3 is trained and deployed, the proposed representation can be plugged on top of FR3 to obtain additional performance increase. Fig.5a shows two bars for every distortion. The left bar in each distortion indicates the performance of model ResNet-18 trained only on undistorted data using first-order representation (in blue) and $f(x)$'s corresponding increase for RR (in red) on level 5 distortions. The right bar shows model FR3's first-order results in blue and it's corresponding increase (in red) when converted to the proposed second-order representation. The "plug-in" functionality is not unique to ResNet-18 and works on multiple networks. Fig.5b shows other common networks (x-axis) all of whose averaged original results (in blue) across levels and distortions are improved using NR (in green) and additionally by RR (in red). From the results, a NR ResNet-18 is as robust as ResNet-101.

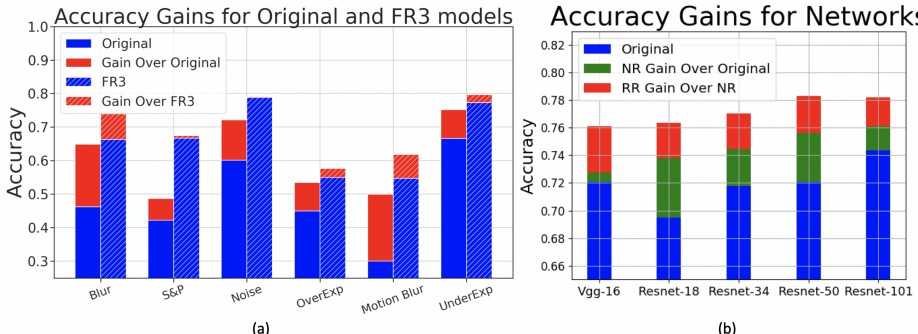

Figure 5: Robustness of accuracy gains of using second-order representation over first order representation.

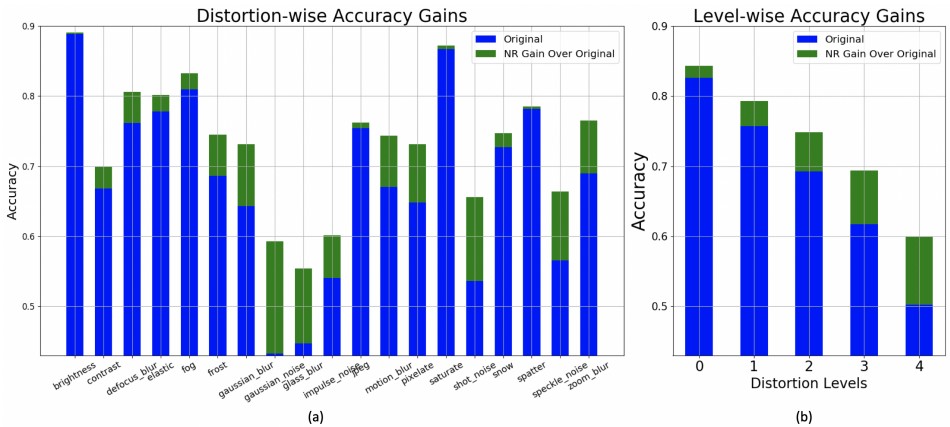

Figure 6: Robustness of accuracy gains of using second-order representation over first order representation on 19 distortions in CIFAR-10-C.

**Robustness on complex distortions:** The proposed representation is validated on CIFAR-10-C (Hendrycks & Dietterich, 2019). This dataset is corrupted by 19 distortions. ResNet-18 trained on undistorted CIFAR-10 images is used as baseline original architecture. The averaged accuracy across 5 levels of progressive distortion on all 19 distortions is shown in Fig. 6a. The average increase in accuracy for proposed second order NR representation over all distortions and distortion levels is 3.54%. In 7 of the 19 distortions, there is an accuracy improvement of 5% or more. Only in brightness and saturation distortions, the average accuracy of proposed NR decreases by 0.23% and 0.59% respectively. The level-wise accuracy gain of proposed second-order representation over original first-order representation averaged over all distortions is visualized in Fig. 6b. The proposed method provides just under 10% increase during severe distortion. Note that since there are no distorted images, there are no statistics to be measured and used to obtain RR representation. Further analysis and distortion-level plots are presented in Appendix A.5.

**Robustness on higher resolution images:** The proposed approach is implemented on higher resolution images of size $96 \times 96 \times 3$ in STL-10 dataset (Coates et al., 2011). ResNet-18 architecture is adopted with an extra linear layer to account for change in resolution. STL-10 is corrupted by the 6 distortions from Sec. 5.1. The results of both NR and RR approaches increase by an average of 2.56% in all but underexposure distortion. In underexposure, the accuracy drops by 1.05%. In level 5 of both blur categories, the RR increase is 6.89%. Additional results are provided in Appendix A.4.

## 5.3 MULTI-DIRECTIONAL FEATURE COST ENHANCEMENT

As indicated in Sec. 3 the dimensionality and time cost for generation of $r_x$ may become prohibitive on large datasets for large networks. We propose two enhancements to mitigate these costs without a substantial loss in performance.

Table 1: Accuracy comparison : Feature Generation Time Analysis.

|  | Gaussian Blur | Salt and Pepper | Gaussian Noise | Over Exposure | Motion Blur | Under Exposure |
|---|---|---|---|---|---|---|
| $\nabla_x$ RR | 82.96% | 60.94% | 78.99% | 70.34% | 71.22% | 84.95% |
| $\nabla_{f_{L-1}(x)}$ RR | 80.38% | 60.37% | 76.06% | 69.76% | 69.32% | 84.05% |
| $\nabla_x$ NR | 80.31% | 56.55% | 76.88% | 67.68% | 66.84% | 83.76% |
| $\nabla_{f_{L-1}(x)}$ NR | 80.21% | 60.33% | 75.83% | 69.70% | 67.53% | 84.21% |

Table 2: Accuracy comparison : Dimensionality Reduction.

|  | Gaussian Blur | Salt and Pepper | Gaussian Noise | Over Exposure | Motion Blur | Under Exposure |
|---|---|---|---|---|---|---|
| Original $r_x$ | 82.96% | 60.94% | 78.99% | 70.34% | 71.22% | 84.95% |
| $r_x$ dim = 1024 | 82.00% | 58.54% | 77.48% | 69.06% | 69.70% | 84.52% |
| $r_x$ dim = 512 | 82.14% | 58.06% | 77.76% | 69.26% | 69.94% | 84.68% |
| $r_x$ dim = 256 | 82.08% | 57.93% | 77.84% | 68.67% | 69.70% | 84.44% |

**Feature generation time analysis :** The proposed representation requires generating adversarial images over all possible $N$ classes during testing phase. Generating $r_x$ in this fashion takes around 0.28s for every image on VGG-16. The reason for the large cost is the multiple backpropagation that happen over all 16 feature extraction modules to create adversarial noise in $x$ domain. However, the generation of perceptually similar adversarial images in the spatial $x$ domain is not our goal. Instead of generating adversarial images as $x + \epsilon \text{sign}(\nabla_x J(W, x, i))$, we generate adversarial data points in the $f_{L-1}$ domain. That is, instead of backpropagating to $x$ and taking the partial derivative $\nabla_x J(W, x, i))$ w.r.t $x$, we backpropagate only one layer to $f_{L-1}$ and obtain the partial derivative w.r.t $f_{L-1}(x)$. Hence, in place of adversarial images, we create adversarial data points in $f_{L-1}(x)$ to obtain $r_x$. Generating $r_x$ in this fashion takes 0.0056s for every image. The average accuracies across levels for all six distortions is presented in Table 1. The performance of the $\nabla_{f_{L-1}(x)}$ RR and NR representations are comparable to their $\nabla_x$ counterparts among all distortions. Additionally, $r_x$ generation for ResNet-18 and ResNet-101 in Fig.5b takes 0.4 and 0.98s respectively. However, creating advesarial datapoints and obtaining $r_x$ takes 0.042 and 0.036s respectively. Hence, this alternate feature generation can be used on networks with large feature extraction modules.

**Dimensionality Reduction:** The dimensionality of the second order representation $r_x$ is $(N * d_{L-1}) \times 1$. However, this feature is derived from a representation space where a linear classifier can discriminate between $N$ classes. On such a representation space, the application of simple dimensionality reduction techniques yield subspaces that are themselves discriminative. Here, we use PCA on $r_x$ derived from VGG-16 from Section 5.2. The original $r_x$ from VGG-16 is a vector of length $5120 \times 1$. PCA algorithm is applied on all $r_x$. From 5120, we decrease the dimensionality to 1024, 512, and 256. The distortion level averaged results of RR method show only a small decrease in performance displayed in Table 2.

## 6 CONCLUSION

In this paper, we analyzed the representations of deep learning methods from the perspective of point processes. Our construction of a second-order representation space is analogous to existing representations. Features in the second-order representation space are based on relative positioning between data points and trained decision boundaries. Such features were utilized to reduce the vulnerability of deep learning methods to distortions and real-world challenging conditions. Validation was conducted on 5 architectures, 19 distortions and 2 image resolutions. The proposed second-order representation, which is a *plug-in approach*, enhances robustness performance in neural networks.

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

# A    APPENDIX

## A.1    STABILITY OF PROPOSED REPRESENTATION VS FIRST-ORDER REPRESENTATION

In this section, we compare stability of first-order and second-order representations under distortion. In Azulay & Weiss (2018), the authors demonstrate the instability of neural nets. They do so by translating an image by a few pixels, and showing that the network output and the hidden layer representation changes drastically. We show that this instability holds for distortions as well in Fig.7. Borrowing the notations from Sec.1, the pristine image from MNIST dataset is subjected to five levels of progressive blurring all of which are visualized in Fig.7a. Fig.7b shows the difference in the first-order point representation of $f_{L-1}(x_{BlurLvl})$ to the original $f_{L-1}(x_{orig})$ i.e. each of the maps shown in Fig.7b is the $abs(f_{L-1}(x_{BlurLvl}) - f_{L-1}(x_{orig}))$ for the corresponding image in Fig.7a. Ideally, every map in Fig.7b must be zero, or in this case, must be visualized in blue. However, as can be seen, the representation has values nearing 1 (orange). The representation also varies significantly across blur levels.

We extract the second-order features from the distortion-free image as well as the blurry images using Eq. 2. We then obtain the difference between the $r_x$ of original distortion-free image and the blurred images, which are visualized in Fig.7d. Perceptually, the relative difference between blur and original maps are less in proposed representation (more pixels are blue compared to orange) in Fig. 7d compared to the first-order representation in Fig.7b. The t-SNE representations from Fig. 2 are also visualized in Figs. 7c and e.

## A.2    LOSS FUNCTIONS FOR MULTI-DIRECTIONAL FEATURE GENERATION

In Sec.3, we describe the generation procedure for the multi-directional feature $r_x$. The feature is a concatenation of multiple gradient directions each of which is generated as $r_i = \sum_{j=0}^{k-1} abs(\nabla_{W_L^i} J(W, x_j, i))$. Here $J$ is the cost function that is being backpropagated. Generally, neural nets define loss functions that indicate the empirical difference between the predicted and required values. These loss functions control the amount of required backpropagation and consequently the gradients. Since gradients are our features, the choice of loss functions changes the proposed second-order representations. We compared multiple loss functions all of whose distortion-wise level-wise averaged RR results are provided in Table 3. CE is Cross Entropy, MSE is Mean Squared Error, L1 is Manhattan distance, Smooth L1 is the leaky extension of Manhattan distance, BCE is Binary Cross Entropy and FL is Focal Loss. Notice that the results of all loss functions exceed that of the original. The results of MSE, L1 and Smooth L1 are presented by backpropagating a one-hot vector (with 1 at $i$) multiplied by the average of all activations in the train non-distorted dataset. For ResNet-18, this number is close to 11.

From Table 3, linear or near linear loss functions like L1 and MSE perform better than log based losses. This is because in the $f_{L-1}$ representation space where a linear classifier can discriminate

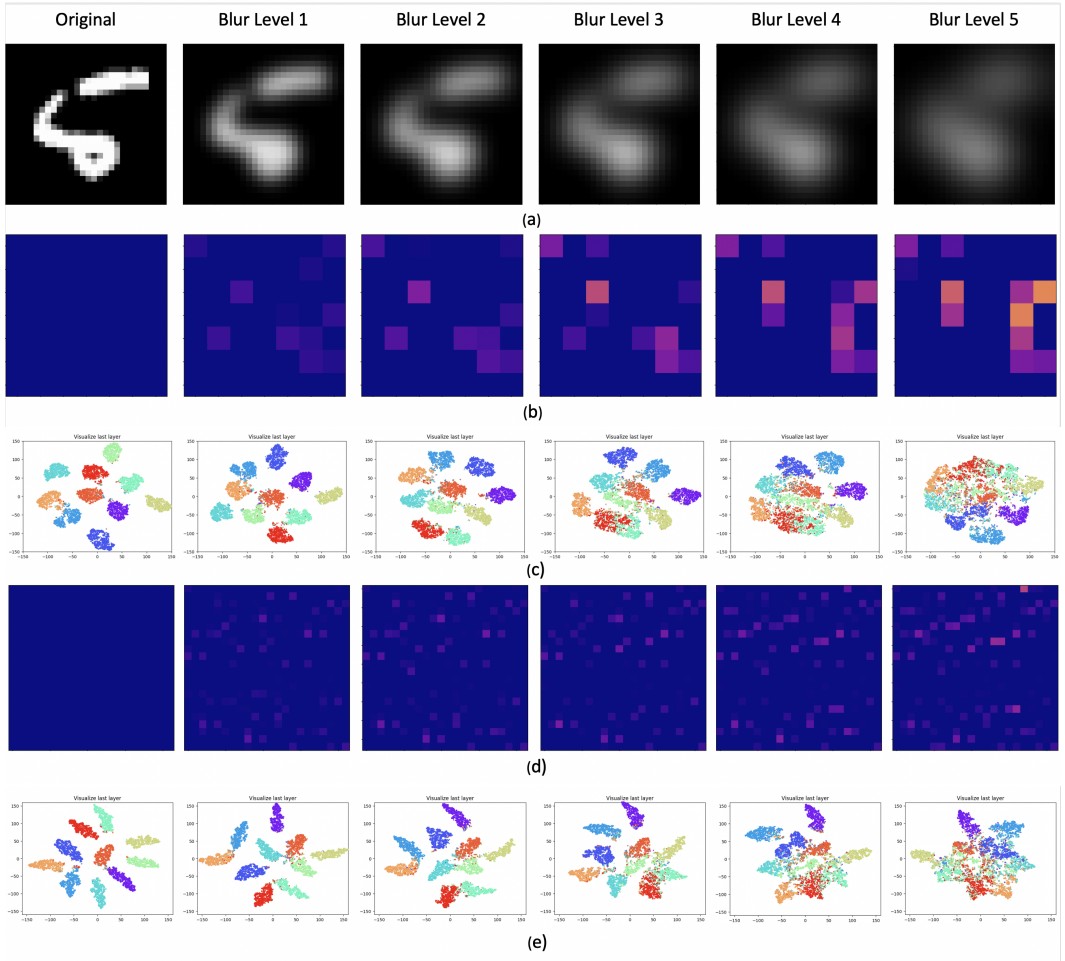

Figure 7: (a) Progressive blurring of a sample image from MNIST dataset (b) Difference map between original image activation in $(L-1)$ layer and corresponding blur activation map from the same layer. Ideally they should be zero (or blue). Orange represents one (c) t-SNE maps of corresponding blurry MNIST testset for first order representation (d) Difference map for proposed representation. More pixels in blurry image match the original, and hence more blue pixels (e) t-SNE maps of corresponding blurry MNIST testset for proposed representation.

Table 3: Proposed RR accuracies with different loss function

|    | Original | CE | MSE | L1 | Smooth L1 | BCE | FL(gamma=2) |
|----|----------|------|------|------|-----------|------|-------------|
| RR | 67.65%   | 71.99% | 75.38% | 75.12% | 73.51% | 74.23% | 72.91% |

between classes, a linear loss function provides a more accurate distance between data points and decision boundaries. We plot the results of the MSE, L1, BCE and original loss functions for all 6 distortions in Fig. 8. MSE outperforms L1 by $0.26\%$ average accuracy and is used in our experiments.

### A.3 MNIST PERFORMANCE

We consider gaussian blur, and salt and pepper noise as the distortions to evaluate our method on MNIST dataset. These were chosen because of their large impact on performance of classifiers trained on original images. Blur is introduced using a standard gaussian kernel with $0$ mean and progressively worsening standard deviation from $[1.5, 3.5]$ with an interval of $0.5$. This produces five levels of distortion. Sample Gaussian blurred images are shown in Fig.2a. Salt and pepper noise

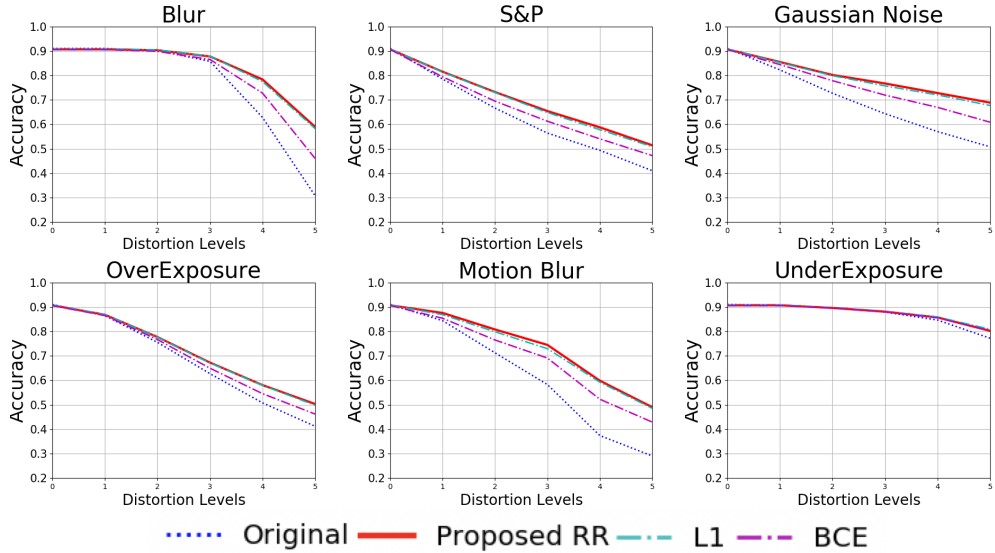

Figure 8: Result comparison for different loss functions. Proposed RR is MSE loss.

is a high frequency noise that occurs in images due to sudden signal disturbances. Salt and Pepper noise is modelled as randomly scattered white pixels in the image with a density that changes from $[0.01, 0.05]$ over five intervals leading to $5$ distorted levels. The code for the noise generation process for both MNIST and CIFAR-10 used in Sec. 5 will be made public. The original network $f_L$ used to obtain second order features $r_x$ is a standard 2-conv-2-FC network. $g_L$ is trained with 3 layers.

The results are presented in Fig.9. The x-axis and y-axis represents the distortion levels and their corresponding accuracy respectively. Level $0$ on the x-axis indicates no distortion and level $5$ indicates the highest amount of distortion. In the plots, original refers to the network trained on undistorted images. It can be seen that our second-order representation based RR and NR methods do not suffer from any drop in accuracy on original first-order undistorted data.

In both the distortions, the proposed NR second-order representation outperforms original method. The performance gain is particularly apparent at higher distortion levels. For blur level $5$, the proposed NR method outperforms the original network by nearly $18\%$. This is a direct indication of the potential of second-order representations over first-order methods. The proposed RR method outperforms the proposed NR and original methods. In blur level $5$, it achieves roughly $30\%$ gain over original network.

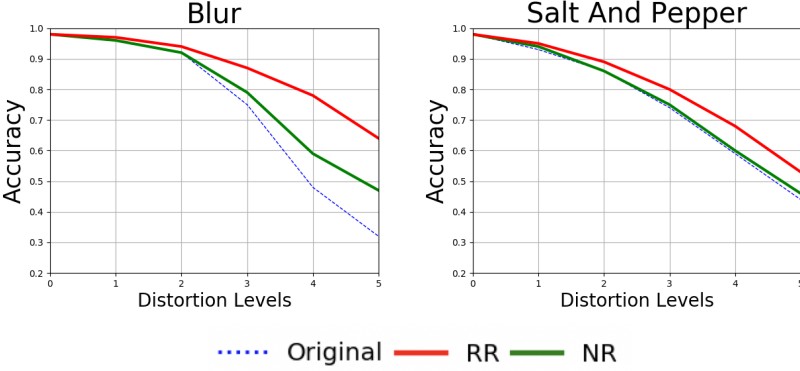

Figure 9: Results on MNIST. NR refers to proposed second-order representation and RR refers to statistics enhanced second-order representation.

## A.4 COMPARISON AGAINST OTHER SECOND-ORDER REPRESENTATIONS

The results in Sec.5.1 indicate that the proposed proxy second-order representations perform better than their true second-order counterpart. ResNet-18 is the base original model used in these experiments. We provide plots for the proposed NR and FR representations when compared against its original first-order representation and kNN classifier with 4 different values of k - 5, 25, 50, 500 in Fig. 10. Note that both our representation and that of kNN are constructed on $f_{L-1}$ representation space of ResNet-18. The results on all 6 distortions are shown. While the first-order original representation lags behind our proposed representation, it performs similar to, or in some cases outperforms, kNN methods. The parameter $k$ does not create a noticeable difference in the results. Even when $k$ was parameterized to a high value of $5000$, the overall results were less than the original.

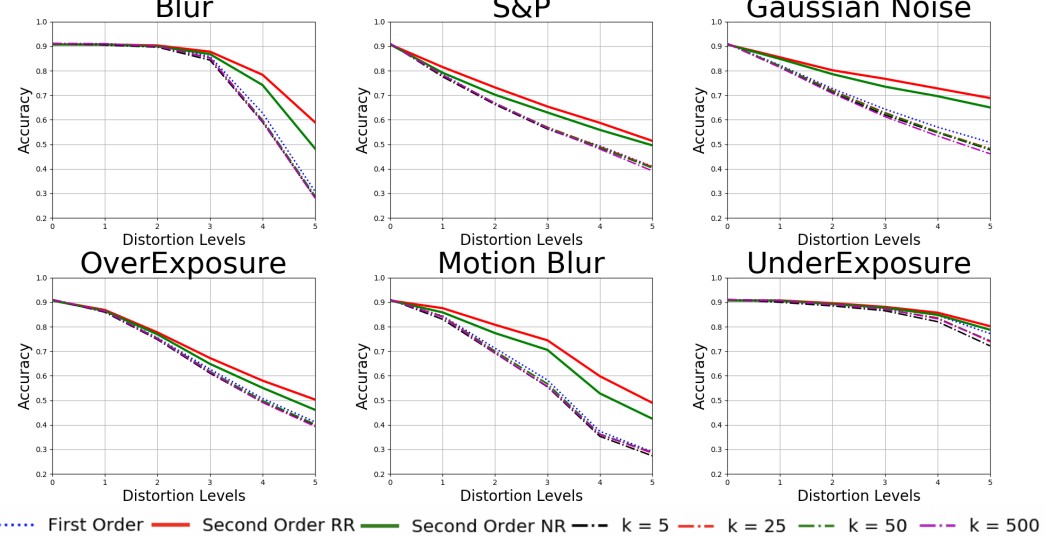

Figure 10: Results of proposed representations compared against kNN classifier.

## A.5 RESULTS ON CIFAR-10-C

The distortion-wise and level-wise results of applying our second-order representation on original ResNet-18 was shown in Sec.5.2. Here we show plots for level-wise results for all 19 distortions in Fig.11. The original trained model is ResNet-18 whose results are presented in blue. Its corresponding second-order representation is shown as NR in green. As mentioned in Sec.5.2, CIFAR-10-C does not provide access to either training distorted images nor their statistics and hence direct RR representation cannot be obtained. However, we calculate mean and standard deviation from a subset of test images and use those for normalization. Notice that doing so converts our RR framework into a Domain Adaptation framework. The results of such a RR framework are provided in red in Fig.11.

CIFAR-10-C consists of a number of real-world complex scenarios. These complex scenarios range from simple gaussian blur and gaussian noise to real-life challenges like defocus blur, motion blur, rain, fog, and frost among others. The accuracy of the original ResNet-18 model trained with pristine images and tested on distorted images range between $[0.9, 0.2]$. Distortions like contrast, Gaussian blur, Gaussian noise, impulse noise, and speckle noise traverse through this entire range and provide ideal means to compare proposed second-order representation against first-order representations. In each of these distortions except contrast, both the NR and RR representations outperforms the original by considerable margins. Contrast is a distortion which even the human visual system cannot readily overcome (Geirhos et al., 2018). In $8$ of the $19$ distortions, the RR representation obtains results that are greater than $10\%$ averaged over all levels. These include Gaussian blur, Gaussian noise, glass blur, motion blur, pixelate, shot noise, speckle noise, and zoom blur. The highest increase is $14.18\%$ for shot noise. In $2$ of the distortions, brightness and saturate, the results decrease by less than $0.6\%$ averaged over all levels. Brightness and saturate both span representation spaces where the gradients to multiple decision boundaries are all very large and are not discriminative enough

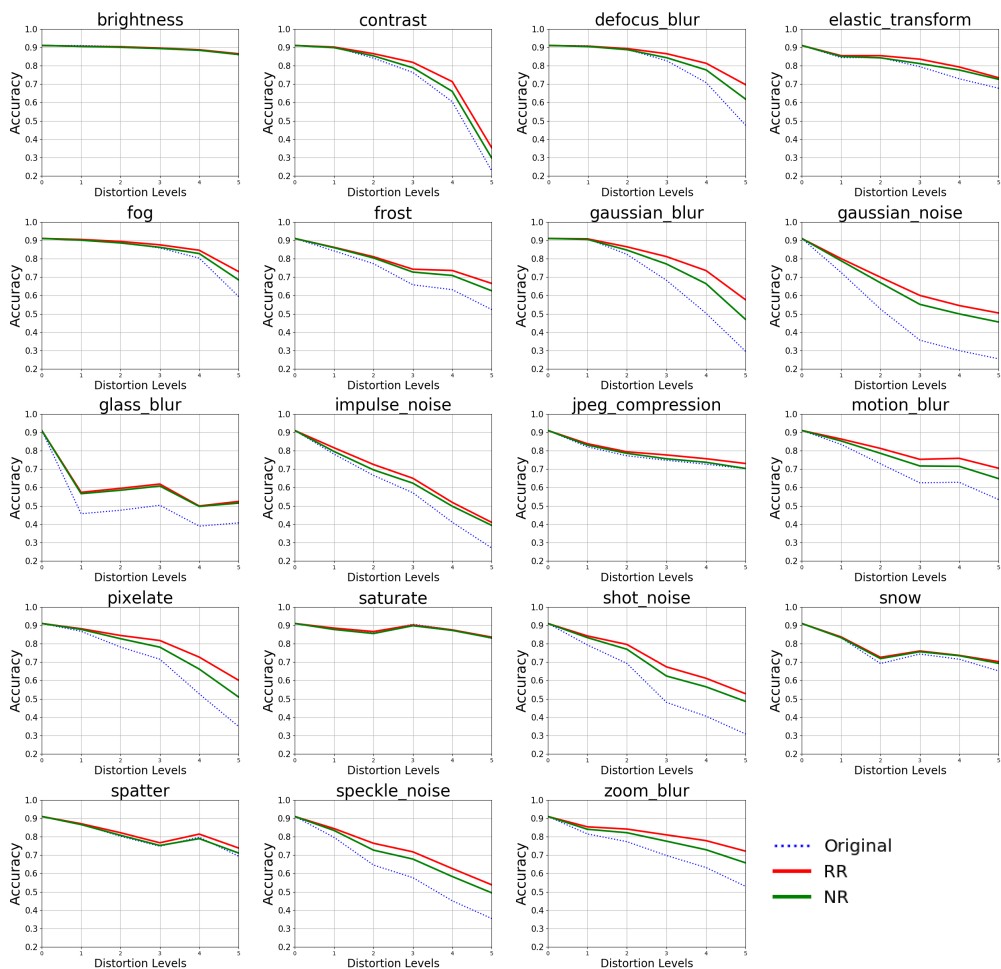

Figure 11: Results of proposed representations on CIFAR-10-C images.

to differentiate between classes. The same explanation also extends to contrast, and fog where the increase is roughly 2% from the original.

Another way of characterizing results is by categorizing distortions by their local and global statistics. Distortions like saturate, brightness, contrast, fog, and frost change the low level statistics in the image domain. Neural networks are actively trained to ignore such changes so that their effects are not propagated after the first few layers. Hence any second-order representation created on the final fully connected layer is going to be analogous to its first-order counterpart. Therefore, the results of the proposed approach and the original model follow each other closely. This provides the motivation to consider creating second-order representations in the first few layers as well.

### A.6 RESULTS ON HIGHER RESOLUTION IMAGES AND ALTERNATE APPLICATIONS

The results of the proposed representation when applied on larger resolution images of STL-10 dataset are provided in Sec.5.2. Here, we show the plots of the proposed representation against its first-order representation on 6 distortions in Fig.12. The increase in distortion-averaged level-averaged results is 2.56% in all except underexposure distortion. While this shows that the second-order representation outperforms its first-order counterpart, the increase in results is not as substantial as in CIFAR-10 and CIFAR-10-C. This is because the original ResNet-18 model was trained on only 5000 training images. In such an under-trained scenario, the cumulative gradients of a data point to all decision boundaries become large and not as discriminative as a well trained architecture. This experiment indicates a limit to the performance increase of the second-order representation as a plug-in on top of

first-order methods. However, it provides insight into using this representation as a control parameter during training procedures to provide an alternate and analogous view of under-trained, well-trained, and over-trained architectures.

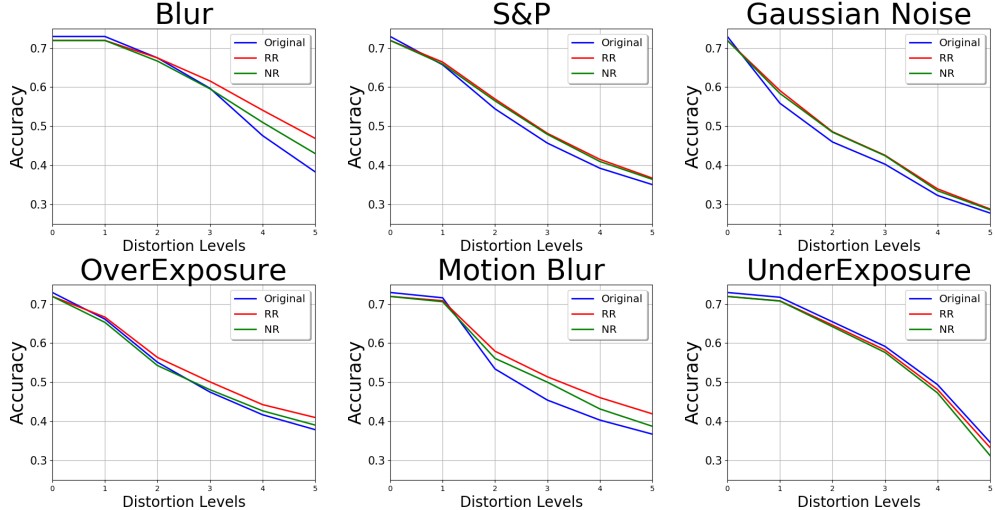

Figure 12: Results of proposed representations on higher resolution STL images.

