# OpenReview forum: "Leveraging Adversarial Examples to Obtain Robust Second-Order Representations"
_ICLR.cc/2020/Conference — Reject_

### Official Review · AnonReviewer1 · 2019-10-24
**Official Blind Review #1**

**Rating:** 3

**Review:**

This paper proposed a method to represent data with its second-order information in the deep network, to improve its representation robustness. To obtain this representation, the authors used artificial adversarial examples to obtain the robust representations. Experimental results show that the new representation is more robust against real-world distortions.

Here are more additional comments on the technical details:
1) The authors proposed a new representation of a datapoint x in equation (2), using the concept of adversarial noise. However, it is still not clear to me how the expression in (2) is derived mathematically. For example, where does the absolute operator in (2) come from? It would be nice if the authors can provide some mathematical derivation or justification of the chose expression, in addition to the empirical evidence.

2) The robust representation is proposed under a discriminative deep network, with predefined class information. Is it possible to extend the work to unsupervised settings (e.g., deep autoencoders), so that the representation robustness is not with respect to a specific definition of class labels?

3) As the authors mentioned in Section 5.3, the computational cost is high for generating such robust representation. Is there any approximation procedures which can be taken to speed up the proposed robust feature representation?

**Experience Assessment:**

I do not know much about this area.

**Review Assessment: Checking Correctness Of Derivations And Theory:**

I assessed the sensibility of the derivations and theory.

**Review Assessment: Checking Correctness Of Experiments:**

I assessed the sensibility of the experiments.

**Review Assessment: Thoroughness In Paper Reading:**

I read the paper at least twice and used my best judgement in assessing the paper.

---

### Official Review · AnonReviewer3 · 2019-10-25
**Official Blind Review #3**

**Rating:** 1

**Review:**

This paper presents a method to represent input points by their relative positions to decisions boundaries using an adversarial attack instead of the more usual output of a deep model (before the last linear classification layer). The motivation of the method is to obtain representations that are more robust to input distortions such as blur, noise, over-exposure, etc. More specifically, the representation obtained by a deep model before the last linear classification layer (f_L) is replaced by a concatenation of the perturbation needed for that representation to be misclassified by f_L. Using this new representation, they are able to obtain more robust classifiers against input perturbations.

This paper should be rejected for several reasons. First, the motivation of the paper is unclear; while being more robust to input perturbations is a desirable property of ML models, the reason to use these second-order representations is not justified. Second, the paper contains several imprecisions about the methodology, which make it complicated to follow. Finally, the presented results are not very convincing; it seems that most of the improvement obtained comes from deriving a larger representation of each sample and using a MLP to classify it instead of a simple linear layer.

Detailed arguments:

It is not clear how r_i is obtained in section 3. This becomes even more confusing in section 5.3, when the features are instead generated as the gradient of the loss with respect to f_{L-1}. To the best of my understanding, r_i is supposed to be the distance between a point (before the final layer) to the decision boundary for class i. If we have x_k, the input which yields the closest point to the decision boundary of class i (as depicted in Fig 3.(b)), then we should have $r_i = f_{L-1}(x) - f_{L-1}(x_k)$. However, r_i is defined as $\sum_{j=0}^{k-1} \abs(\nabla_{W_L^i} J(W, x_j, i)$. In this formula, the absolute value clearly changes the direction of the gradient. It would be appreciated to have a clear explanation for the reason of the absolute value.

The paper contains incorrect statements about standard adversarial attacks. FGSM is a one-step attack, and it does not make sense to do several iterations of it. A more powerful multi-step attack that can be considered is PGD.

A lot of emphasis is put on the fact that the obtained representations might be too large to scale well to datasets where there are more classes than the ones considered in the article. However, there are no experiments to support this - the experiments in the paper do not go beyond 10 classes.

Additional comments:

Some citations need to be fixed, removed or changed:
Second paragraph of the introduction: “[...] can discriminate between classes (Goodfellow et al., 2016)(Krizhevsky et al., 2012)(He et al., 2016).” Deep neural nets were not initially proposed in these papers, so it is somehow inappropriate to cite these papers here (except, maybe, Krizhevsky'12).

Third paragraph of the introduction: “[...] the representations (fL−1) are linearly projected (Wang et al., 2019).” Same here.

The main figures to explain the idea are not really helping the argument: Figure 1. (b) is simply Fig 1. (a) rotated 180 degrees and re-scaled, but the information is essentially the same; maybe not the best example.

Figure 2: the provided t-SNE projections do not show a clear advantage of the proposed method. A quantitative analysis would better support the argument.

The time measurement given in section 3 is not indicative: what are the 320s? An indication in terms of epochs or training iterations would be better.

It would be appreciated to have measures of standard deviations for the results especially in tables 1 and 2 since some results are very close and might depend mostly on random perturbations. Moreover, it is not mentioned for which datasets these tables are for.


**Experience Assessment:**

I have published one or two papers in this area.

**Review Assessment: Checking Correctness Of Derivations And Theory:**

I carefully checked the derivations and theory.

**Review Assessment: Checking Correctness Of Experiments:**

I carefully checked the experiments.

**Review Assessment: Thoroughness In Paper Reading:**

I read the paper thoroughly.

---

### Official Review · AnonReviewer4 · 2019-11-03
**Official Blind Review #4**

**Rating:** 1

**Review:**

The authors propose an approach to reduce the sensitivity of neural networks to visual distortions. To do so, they modify the representation of a data point within a network, using its relative position (to other points) in representation space rather than its absolute position  (which is measured as distance of a point to the decision boundary of each class). They then evaluate their approach on common corruptions from the CIFAR-C dataset.

The paper addresses an important problem, however I do not find the high-level motivation behind the proposed approach or the experimental results sufficiently convincing.

In particular, conceptually, it is completely unclear why the second order representations should be more resilient to visual distortions. Even in the sample illustration provided in Figure 2, it is evident that the clusters in this new representation space are not invariant to visual distortion, or even are significantly more invariant than the first-order representation. At a more fundamental level, given that the accuracy of the original network does drop, it is tautological that the distance of a point to the decision boundaries is decreasing under distortion (and thus is sensitive to it).

Experimentally, my chief concerns are:
1. When the authors evaluate the proposed second-order representations, they use networks with additional layers which do not seem to be present in the original baseline. Prior work [Hendrycks and Dietterich, 2019] has shown that model capacity has a marked influence on the corruption robustness of a network. Thus, it is unclear where the improvement here is coming solely from the additional capacity compared to the original model. The baseline network that the authors compare to should also include the additional layers.
2. The results reported for some of the baselines seem inconsistent with prior work.
a) For example, the authors state that the results for the Hossain et al. [2018] baseline are similar to NR1, which is worse than the original network. However, Hossain et al. report an improvement for the same corruptions and the same dataset in their paper. Where is this inconsistency coming from? Do the authors train with the DCT filtering or is it only applied at test time? If it is the latter, it could explain why the authors fail to reproduce the baseline correctly.
b) For the adversarially robust network baseline, why did the authors choose an FGSM adversary? FGSM is not typically used to train state-of-the-art robust models because of the existence of much stronger attacks (such as PGD). How was the eps used to train the network chosen? In prior work, Kang et al. [2019; arxiv:1908.0801] evaluate both L2 and Linf robust models and show improvements over the baseline for several common corruptions. This seems to suggest that the robust model baseline reported in this paper is not accurate/representative.
3. Moreover, given that the representation size scales with number of classes, the proposed method should be evaluated on datasets with more classes such as CIFAR-100 or ImageNet. Improvements demonstrated in these settings with dimensionality reduction would be more convincing.

Overall, my main reservations are: a) the lack of a conceptual justification for the proposed approach, and b) issues with the experimental evaluation, particularly in the reported baselines and how they seem to contradict prior work. Thus, I recommend rejection.

Other comments:
- The right-side of Figure 1 is essentially the same map flipped.
- Why was the MSE loss chosen for J? Given that the network was presumably trained with cross-entropy, this choice seems somewhat arbitrary. Are the results consistent for cross entropy loss as well? The authors should include these results in the appendix.
- Figure 4 is very hard to read---the authors should change the plotting style to make the results more legible.
- For the results in Figure 5, the authors should once again compare to adding extra layers to the baseline networks as well.
- In Table 1, why does the performance of NR improve when the features are based only on gradients for the last layer?

**Experience Assessment:**

I have published in this field for several years.

**Review Assessment: Checking Correctness Of Derivations And Theory:**

N/A

**Review Assessment: Checking Correctness Of Experiments:**

I assessed the sensibility of the experiments.

**Review Assessment: Thoroughness In Paper Reading:**

I read the paper at least twice and used my best judgement in assessing the paper.

---

### Decision · Program_Chairs · 2019-12-19

**Decision:**

Reject

**Comment:**

The authors propose a method to train a neural network that is robust to visual distortions of the input image. The reviewers agree that the paper lacks justification of the proposed method and experimental evidence of its performance.